# CHECKEMBED: EFFECTIVE VERIFICATION OF LLM SOLUTIONS TO OPEN-ENDED TASKS

## ABSTRACT

Large Language Models (LLMs) are revolutionizing various domains, yet verifying their answers remains a significant challenge, especially for intricate open-ended tasks such as consolidation, summarization, and extraction of knowledge. In this work, we propose CHECKEMBED: an accurate, scalable, and simple LLM verification approach. CHECKEMBED is driven by a straightforward yet powerful idea: in order to compare LLM solutions to one another or to the ground-truth (GT), compare their corresponding answer-level embeddings obtained with a model such as GPT Text Embedding Large. This reduces a complex textual answer to a single embedding, facilitating straightforward, fast, and meaningful verification. We develop a comprehensive verification pipeline implementing the CHECKEMBED methodology. The CHECKEMBED pipeline also comes with metrics for assessing the truthfulness of the LLM answers, such as embedding heatmaps and their summaries. We show how to use these metrics for deploying practical engines that decide whether an LLM answer is satisfactory or not. We apply the pipeline to real-world document analysis tasks, including term extraction and document summarization, showcasing significant improvements in accuracy, cost-effectiveness, and runtime performance compared to existing token-, sentence-, and fact-level schemes such as BERTScore or SelfCheckGPT.

## 1 INTRODUCTION

Large Language Models (LLMs) (Zhao et al., 2024b; Minaee et al., 2024) are transforming the world. One particular ongoing challenge in the LLM design is hallucination detection (Petroni et al., 2019; Huang et al., 2023a; Zhang et al., 2023b) and the corresponding overall verification of LLM answers (Chang et al., 2024; Rawte et al., 2023). Numerous works tried to address this issue, focusing on – for example – grounding knowledge or explainability, and even giving rise to questions regarding methodology and epistemology of artificial intelligence (AI) in general (Fleisher, 2022).

Recent verification methods and their building blocks, such as SelfCheckGPT (Manakul et al., 2023) and BERTScore (Zhang et al., 2020) focus on individual fact checking and token- as well as sentence-level analysis. To achieve this, all these methods have to use some form of *comparison* of two passages of text. This could be comparing an LLM answer to a ground-truth (if available), or comparing two different LLM answers to the same question to determine whether these answers are similar (which implies the LLM is certain of its answer) or different (which implies that the LLM is unsure of what the answer really is). For example, with BERTScore, comparing two passages of text involves computing embeddings of *all* words in each passage, and calculating certain scores for *all pairs* of embeddings from both passages.

However, the problem of verifying LLM answers to more complex tasks, such as open-ended document analyses, still poses a challenge. As an example of such a task, consider extracting legal terms and their definitions from a document. The difficulty of verifying the answers to such a task is due to the inherent lack of structure, even assuming one has the ground-truth answer. Namely, the output of such a request would be a potentially long list of definitions. To verify this answer, existing methods such as SelfCheckGPT or BERTScore would go ahead and compare all pairs of words between different solutions and/or the ground-truth. This is fundamentally infeasible, because their token-, sentence-, and fact-based approaches scale poorly with growing task sizes. Moreover, we observe that while two different LLM answers can comprise of very different sets of sentences, their

*meaning* could indeed be very similar. This aspect is not well reflected by sentence- and token-level schemes, leading to them being inaccurate for such complex tasks.

In this work, we propose CHECKEMBED: an approach for *simple*, *scalable*, and *accurate* verification of LLM solutions to such tasks (**contribution 1**). The **key idea** behind CHECKEMBED is to *obtain and compare embeddings of full LLM answers*, or their sizeable chunks, instead of focusing on individual sentences, facts, or tokens. CHECKEMBED relies on the fact that modern embedding models are highly capable; for example, they can be based on powerful Decoder-only LLMs (Lee et al., 2024). Thus, they provide high-dimensional embeddings that can faithfully reflect the *meaning* of the embedded text. *We harness this observation as a basis for CHECKEMBED.* To motivate this idea and assumption, consider Figure 1. In this figure, we illustrate two *very different* passages of text that still describe the *same* concept, and two *very similar* passages of text that describe two *very different* concepts. Interestingly, the cosine similarities as proposed in CHECKEMBED between the embeddings of two different and two similar passages are – respectively – low and very high, supporting the key idea behind CHECKEMBED. BERTScore and SelfCheckGPT are outperformed by CHECKEMBED in both accuracy and runtime.

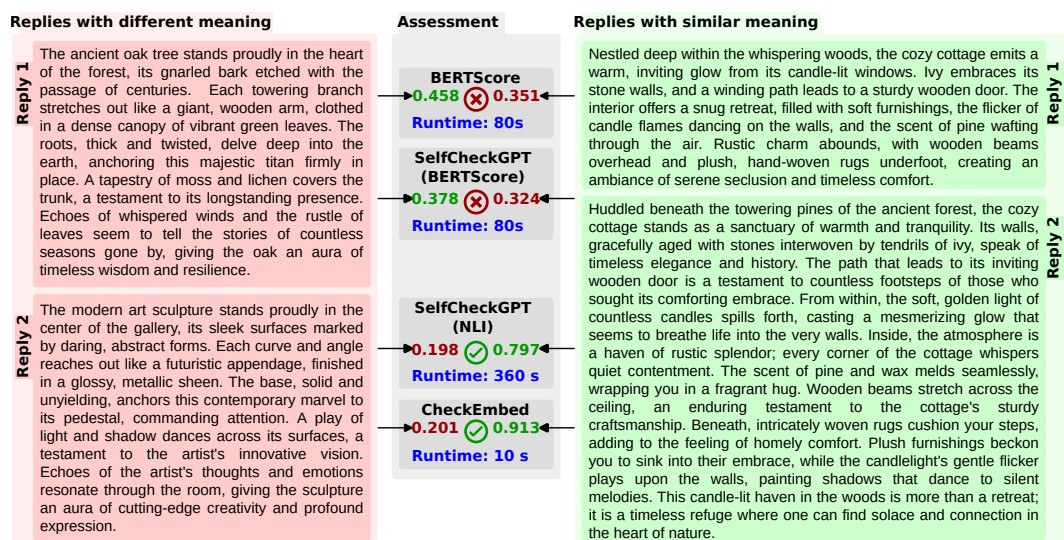

Figure 1: We show two sets of two LLM replies each: Replies explaining different concepts using similar wording (left) and ones explaining similar concepts using different wording (right); the queries used to generate these replies can be found in the Appendix. We compare CHECK-EMBED to two variants of SelfCheckGPT: one that uses BERTScore as a subroutine, and one that harnesses the Natural Language Inference (NLI), which classifies relationships between texts as entailment, neutral, or contradiction, and utilizes a fine-tuned DeBERTa-v3 model (He et al., 2021) to detect textual contradictions by computing a contradiction score based on the logits for 'entailment' and 'contradiction'. We also compare to BERTScore as a standalone baseline. While BERTscore and SelfCheckGPT (BERTScore) assess the semantically unrelated replies as more related than the related ones (because these two baselines have been designed to mostly target the verification of individual sentences or facts), CHECKEMBED *correctly differentiates between semantically related and unrelated replies*, and outperforms SelfCheckGPT (NLI). We use ChatGPT-4o with temperature = 1.0 for replies and the gpt-embedding-large model for generating embeddings.

We design and implement a comprehensive verification pipeline based on CHECKEMBED (**contribution 2**). The pipeline uses the notion of "stability" of the LLM answer, introduced by SelfCheckGPT, as a supporting mechanism. The idea behind "stability" is to prompt an LLM to reply to a given question several times. If the LLM repeatedly outputs the same solution, it means that it has high confidence in its answer and the hallucination risk is low (i.e., high stability of the LLM answers). Contrarily, if there is a large variance in the LLM answers (i.e., low stability of the LLM answers), the risk of hallucinations is high. In CHECKEMBED, we harness this approach for comparing embeddings of *whole LLM answers, or their sizeable chunks*, pairwise to one another, and to the potential ground-truth (GT), if available. Using such answer-level embeddings enables extracting the *meaning* of a given whole reply and to compare it effectively to others and to GT. We show that this strategy is effective and results in embeddings that are close to each other with respect to different distance metrics in cases where the LLM gives correct answers, and with embeddings that are far away, if the LLM is uncertain of the answer or the answer is not of high quality.

As a part of the CHECKEMBED pipeline, we offer assessment metrics that show both how each of the LLM answers compares to any other answer and to the potential GT, and succinct summaries.

The former is provided in the form of embedding heatmaps. The latter are statistical summaries that can be used as user-specified thresholds to drive decision engines in practical deployments on whether a given LLM answer is good enough to be accepted, or not and thus has to be re-generated.

We apply our verification pipeline that implements the CHECKEMBED idea to several real-world use cases in document analysis, namely extracting terms and definitions as well as summarizing documents (**contribution 3**). In addition to the high accuracy, a large advantage of this approach is its *speed* and *simplicity*: all one has to do is to embed the LLM answers and compare them to one another using cosine similarity or other vector distance measures.

We show high advantages in accuracy and runtimes (**contribution 4**). When the ground-truth is available, CHECKEMBED offers closely matching scores for LLM answers. Specifically, we obtain very high scores for high-quality LLM answers and low scores when the LLM answer is a mismatch. This provides an advantage over comparison baselines that often provide mismatching scores.

## 2 THE CHECKEMBED DESIGN & PIPELINE

We now describe the CHECKEMBED pipeline, which is summarized in Figure 2.

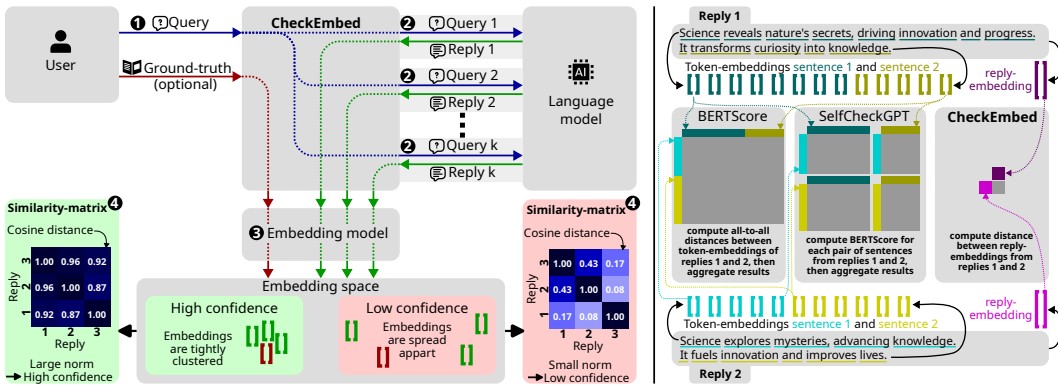

Figure 2: Overview of the CHECKEMBED pipeline (left) and comparison between BERTScore, SelfCheckGPT, and CHECKEMBED (right).

The CHECKEMBED pipeline for verification of LLM's responses consists of the following key parts. First, a user sends a **question ⑦ to the LLM ❶** with all the essential input data. The pipeline enables batching these questions, i.e., it is possible to send multiple questions in the same pipeline and they pass through each of the next stages individually. Next, the pipeline **prompts the LLM ⌘ several times ❷** with the same question ⑦; the user sets this number ($k$). Each reply ⑩ has no prior knowledge of the previous answer guaranteeing that there is no bias. $k$ introduces a tradeoff: more responses (higher $k$) means more compute time and cost (more tokens used), but also a better check of correctness. However, as we show in Section 4, CHECKEMBED enables high level of confidence in its verification outcome even when $k$ is low. The next stage of the pipeline is the **embedding of the answers ❸**. Each reply is embedded, using a pre-specified embedding model (another user input). The potential ground-truth answer ⑩ is also embedded. In the final stage, the **embeddings of the replies are compared pairwise ❹**. We use established metrics, most importantly the cosine similarity; we also experiment with Pearson correlation. Other measures are possible as the pipeline enables seamless integration. The pairwise similarity scores of embeddings are grouped into a (symmetric) heatmap matrix, which is summarized using a selected measure in order to provide a simple threshold number that can be used to drive decision making in practical deployments.

## 3 SCALABILITY ANALYSIS

We provide a brief scalability analysis showing why CHECKEMBED is fundamentally faster than BERTScore and SelfCheckGPT. We denote the number of answers requested from the LLM with $k$. We assume the same dimensionality of all used embeddings and that computing a score of two embeddings is negligible and takes $O(1)$ time (e.g., Numpy supports highly efficient Pearson correlation and cosine similarity). Without loss of generality, we also assume that a single reply or the ground-truth contain $s$ sentences, and each sentence contains $t$ tokens. When comparing the baselines, we consider counts of two most compute intense operations within the pipeline: the number of embeddings to be constructed and the number of similarity operations to be conducted.

In CHECKEMBED, there are $k$ embeddings to construct, and $O(k^2)$ similarity operations to run.

Next, one can apply BERTScore straightforwardly to two passages treated as long sentences, each such passage consists of $st$ tokens. This means $O((st)^2) = O(s^2t^2)$ embedding comparisons have to be performed for any two passages (for each pair of compared sentences, one compares every pair of individual tokens/words), resulting in a total of $O(k^2s^2t^2)$ embedding comparisons as this is done for $O(k^2)$ pairs of LLM answers, and a total of $O(k^2)$ embedding constructions.

Finally, SelfCheckGPT assesses a given LLM reply by comparing it to all sample replies collected. To simplify the following derivations, assume that in an individual comparison of two LLM replies, these replies consist of $s_1$ and $s_2$ sentences, respectively. Now, for each such comparison, Self-CheckGPT uses BERTScore, where the two input passages $x$ and $y$ to BERTScore consist of $s_1s_2$ sentences each, i.e., both passage $x$ and passage $y$ contain all the sentences from its corresponding LLM reply, repeated as many times as the number of sentences in the other LLM reply (this is conducted to enable comparing all sentences from each reply pairwise). This gives (using the above BERTScore formulae) $O(ks^2)$ embedding constructions (there are $k$ LLM replies) and $O(ks^2s^2t^2) = O(ks^4t^2)$ embedding comparisons.

## 4 EVALUATION

We now show the advantages of CHECKEMBED over the state of the art.

**Comparison Baselines** We compare CHECKEMBED to two key baselines, **SelfCheckGPT** and **BERTScore**. SelfCheckGPT comes with **different variants**; we consider the **BERTScore variant** (where BERTScore is used as a subroutine within SelfCheckGPT, and not a standalone method) because of its similarity to our approach, and the **NLI variant**, as it provides a tradeoff between accuracy and cost and comes with top scores.

**Considered Models** First, when prompting the LLM, we explore GPT-3.5, GPT-4, and GPT-4o. Second, when embedding LLM replies, we experiment with different embedding models, namely Salesforce/SFR-Embedding-Mistral (SFR) (Meng et al., 2024), intfloat/e5-mistral-7b-instruct (E5) (Wang et al., 2024b;c), Alibaba-NLP/gte-Qwen1.5-7B-instruct (GTE) (Li et al., 2023b), which all have around 7B parameters, as well as smaller models such as dunzhang/stella_en_1.5B_v5 (STE1.5, 1.5B parameters) (Zhang, 2024a) and dunzhang/stella_en_400M_v5 (STE400, 400M parameters) (Zhang, 2024b). We also use an API-based GPT Text Embedding Large (GPT) model (Zhuang et al., 2024). For BERTScore and SelfCheckGPT, we use the best possible models available for these baselines (i.e., microsoft/deberta-xlarge-mnli (He et al., 2021) and roberta-large (Liu et al., 2019)). We use the default embedding sizes (listed in the Appendix A.2).

**Considered Similarity Measures** We use cosine similarity and the Pearson correlation score. These two follow the same accuracy patterns, and we only show the data for the cosine similarity. We then use the Frobenius norm to extract a single value from the cosine similarity matrices as well as Spearman's rank correlation coefficient for summarization.

**Considered Datasets** In addition to our own datasets, we use one more benchmark: WikiBio. Specificly, we use a subset of the WikiBio dataset (Lebret et al., 2016) that was modified by Manakul et al. (2023) for their evaluation of SelfCheckGPT. It consists of 238 documents based on Wikipedia articles, that were used to generate samples in which hallucinations were introduced. Each sentence of those samples were manually labeled as either "major inaccurate", "minor inaccurate", or "accurate".

### 4.1 DISTINGUISHING SIMILAR AND DIFFERENT TEXT PASSAGES FAITHFULLY

We start the evaluation by extending the motivating example from Figure 1. Specifically, we analyze whether a given verification method is able to clearly distinguish two passages of text that (1) look similar, but come with very different meanings ("Different replies", see the left side of Figure 1 for an example), as well as (2) look different, but have similar or identical meanings ("Similar replies", see the right side of Figure 1 for an example). The used prompts can be found in the Appendix A.1. The prompt sizes used for these two groups are in the range of 25–250 and 100–200 tokens, respectively. To broaden the analysis, we further consider two subtypes of such passages: "Generic" and "Precise". The former are brief while the latter are rich in detailed information (e.g., "Vintage bike" vs. "Old, rusted bicycle leaning against a weathered fence"). We illustrate the results for these two subtypes in Figures 3a and 3b, respectively.

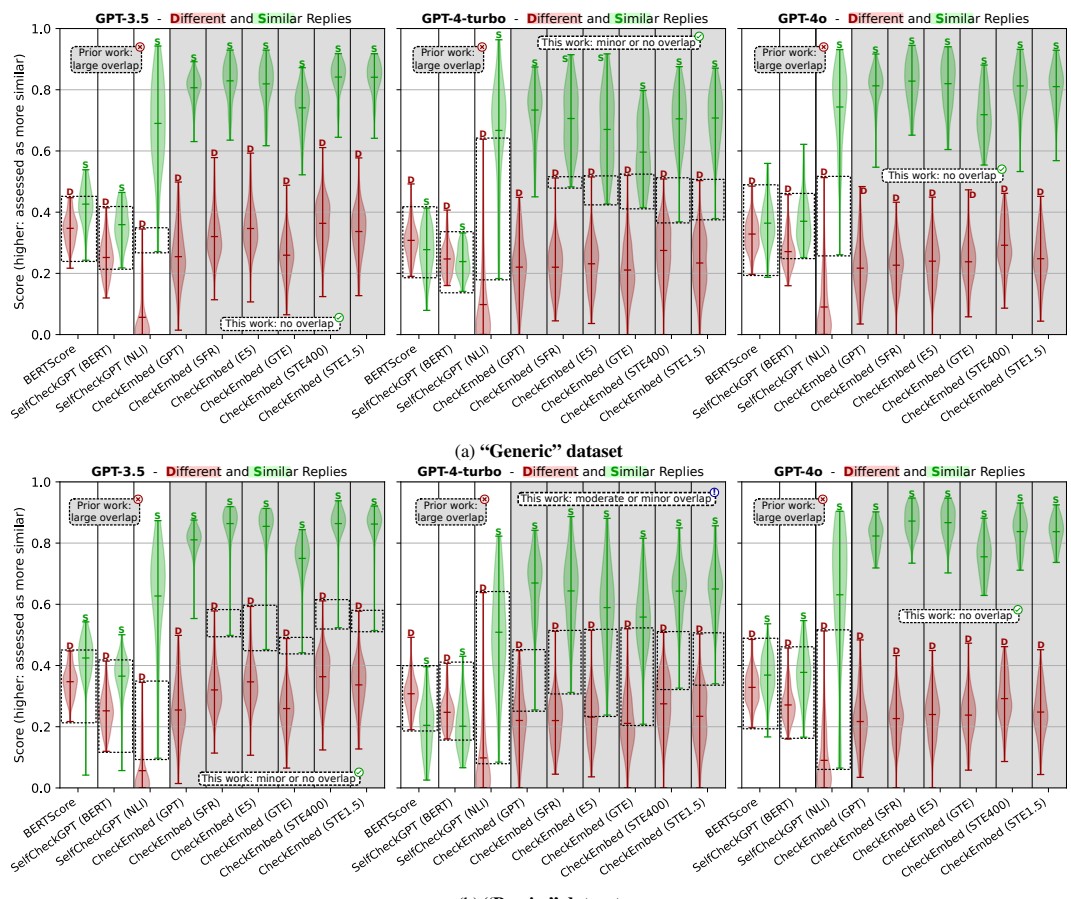

(a) **"Generic" dataset**

(b) **"Precise" dataset**

Figure 3: **Analysis of distinguishing similar and different LLM replies**, details explained in Section 4.1. CHECKEMBED is (highly) effective at appropriately recognizing the similarities and differences in the meaning of the verified text passages. This can be seen from moderate to no overlap between groups of data points corresponding to scores for – respectively – similar and different LLM replies, regardless of the model used. Contrarily, there is a large overlap between these groups of data points for both BERTScore and SelfCheckGPT (BERT), indicating that these baselines perform worse in distinguishing such replies effectively, while SelfCheckGPT (NLI) shows a better, but still noticely inferior to CHECKEMBED, distinction between those two groups.

Importantly, CHECKEMBED comes with *no* (or *very minor*) overlap of scores for similar and different replies. Similar replies come with consistently high similarity scores, while different replies have consistently lower similarity scores. Thus, the **key takeaway** is that CHECKEMBED is highly effective at appropriately recognizing the similarities and differences in the *meaning* of the considered text passages, regardless of their length and style, and also regardless of the harnessed generative and embedding models. Contrarily, both BERTScore and SelfCheckGPT, especially its BERTScore variant, have high overlaps for these passages; thus, CHECKEMBED improves upon the state of the art.

An interesting feature of CHECKEMBED is that, while it *does* distinguish similar and different passages very effectively, it gives *relatively high* scores to the *different* passages; these scores are usually *higher* than the BERTScore or SelfCheckGPT scores for *similar* passages. Despite this, it is still straightforward to distinguish between answers implying similar or different passages, because the CHECKEMBED scores for *similar* passages are *consistently* very high (e.g., with means higher than 0.9 for SFR or E5).

Interestingly, GPT-4-turbo generates replies that are 'the most difficult to distinguish", i.e., it comes with visible (still very low) overlap between similar and different ones, across all embedding models. Contrarily, GPT-4o comes with no overlap whatsoever, while GPT-3.5 has very minor overlap.

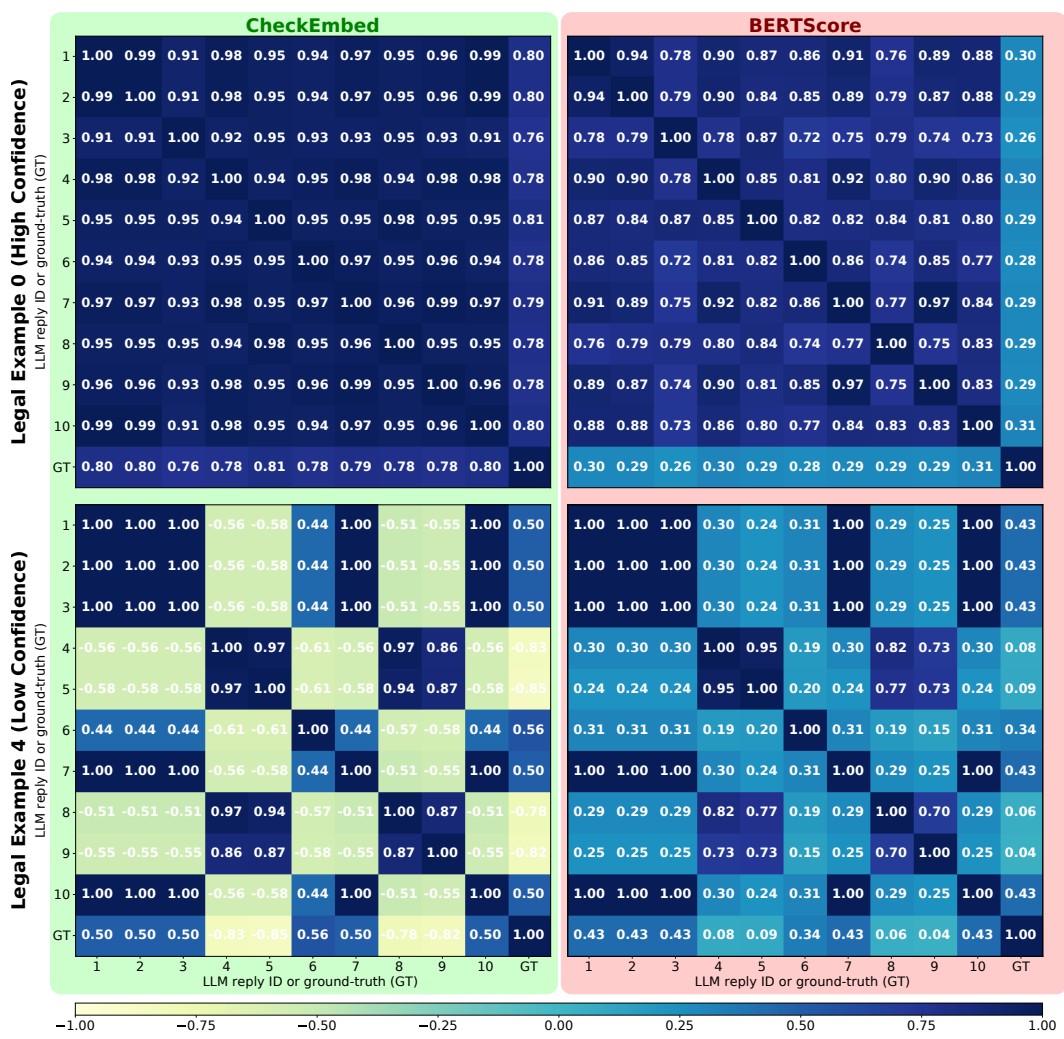

Figure 4: **Analysis of the verification of LLM answers (GPT-4)**, details explained in Section 4.2. We compare to BERTScore; SelfCheckGPT comes with significantly higher runtimes (detailed in Section 4.5) and less competitive scores as it does not focus on open-ended answer-level analysis. The results form a heatmap of the CHECKEMBED's, or BERTScore's, cosine similarity between all LLM replies, and between each reply and the human expert prepared ground-truth (GT). Rows correspond to two representative legal documents, that come with – respectively – high and low LLM confidence in its replies. Embedding model used in both rows: GPT Text Embedding Large.

## 4.2 VERIFYING LLM ANSWERS EFFECTIVELY

Next, we illustrate how CHECKEMBED enables effective verification of LLM answers. As a use case, we consider extracting terms and their definitions from legal documents; the used data is real and it comes from an in-house legal analytics project. In this use case, a prompt to the LLM consists of the contents of a legal document (e.g., an NDA), as well as a request to extract respective terms and their definitions. The prompts can also be found in the Appendix A.1. The prompt sizes used in this task are in the range of 25–600 tokens (we split the documents into chunks as whole documents are often very long and come with total token counts that significantly exceed the recommended maximal sizes for the input of the used embedding models). CHECKEMBED asks the LLM to generate 10 replies ($k = 10$). We illustrate the results for GPT-4 in combination with GPT Text Embedding Large in Figure 4 with additional results presented in Appendix A.4.1. Each figure shows the cosine similarity between all respective LLM replies, and also between each reply and the ground-truth (GT) reply that has been prepared by a human expert.

The results illustrate that whenever CHECKEMBED has very high confidence in its answer (top row in Figure 4), which is visible by consistently having very high similarities between different replies, it corresponds to very high similarity scores between the LLM replies and the ground-truth.

This is the case for all the considered models. Other baselines show mixed results for individual replies, and low similarities between their replies and GT. It shows that, whenever CHECKEMBED has high confidence it the LLM replies, there is high likelihood that these replies are close to the corresponding GT.

In the bottom row of the figure, we provide example results where CHECKEMBED indicates low or mixed LLM's confidence. While many scores are still high (e.g., 0.97), many are much lower, even negative. We manually verified that these particularly low individual scores correspond to LLM replies of very low quality (e.g., only a single term with its definition has been extracted). The low scores overall indicate model's low confidence, which is further supported by corresponding low similarity scores to GT. Here, BERTScore also has low confidence – overall, its scores have a smaller ranger than those of CHECKEMBED, but its relative drop in similarity to GT is similarly as low as that of CHECKEMBED.

Note that the results in the heatmaps directly correspond to the results from Section 4.1 and Figures 3a and 3b in that very high CHECKEMBED scores (e.g., 0.9) indicate high confidence while scores that are lower consistently mean low LLM's confidence.

A useful simple CHECKEMBED measure that indicates the low quality of the LLM answer is a selected summarization measure for a heatmap, for example mean or a matrix norm combined with a standard deviation (std). Whenever the mean is *very high* (e.g., >0.9) and the std is *low* (e.g., <0.05), the answer is of high quality with very high likelihood. Otherwise, one may want to investigate a given situation in more detail. For example, in the top row (example 0), the LLM is very certain of what the answer is; the mean is 0.95 with very low std of 0.06; BERTScore seems to imply hallucinations with lower scores and even more importantly, an std of 0.18.

## 4.3 ANALYZING WIKIBIO DATASET

Next, we discuss the CHECKEMBED performance on an existing benchmark, WikiBio, used to assess SelfCheckGPT (Manakul et al., 2023). Their subset consists of 238 documents based on Wikipedia articles with introduced hallucinations. Each sentence of those samples were manually labeled as either "major inaccurate", "minor inaccurate", or "accurate". Consistently with the SelfCheckGPT evaluation by Manakul et al. (2023), we employed a passage scoring system that aggregates sentence scores: assigning 0 for major inaccuracies, 0.5 for minor inaccuracies, and 1 for accurate sentences—before calculating the average score. This construction allows the utilization of Pearson and Spearman correlation scores to reflect a more nuanced output to quantify the extent of hallucination within passages over more simplistic black-and-white approaches.

Table 1: Passage level correlation on the WikiBio-gpt3 dataset using Pearson and Spearman's Rank Correlation

| Method | Pearson | Spearman |
|---|---|---|
| BertScore | 67.7 | 67.9 |
| SelfCheckGPT | | |
| w/ BERTScore | 57.4 | 54.6 |
| w/ NLI | **74.1** | 73.8 |
| **CheckEmbed** | | |
| w/ GPT | 66.8 | 72.6 |
| w/ STE400 | 68.5 | 72.9 |
| w/ STE1.5 | 69.9 | 73.8 |
| w/ E5 | 71.6 | 74.1 |
| w/ SFR | 72.2 | 76.2 |
| w/ GTE | 73.6 | **76.2** |

An overview of the results is in Table 1, with the full results being presented in Appendix A.4.3. CheckEmbed demonstrates robust performance compared to existing baselines, particularly in Spearman's correlation, where its results are significantly higher. For Pearson's correlation, CHECK-EMBED is marginally outperformed by SelfCheckGPT's NLI variant, but it is more than 30× faster to compute.

## 4.4 DETECTING FINE-GRAINED HALLUCINATIONS

While CHECKEMBED is primarily targeted at verification of open-ended tasks, we also investigate whether CHECKEMBED can be used to detect small fine-grained hallucinations, such as mistakes in individual facts. The results are in Figure 5 and 6 and the used prompts can be found in the Appendix A.1. The task analyzed is summarizing scientific and legal articles. For each article considered, we generate a summary with no errors (labeled as "ground truth"), and we also ask the

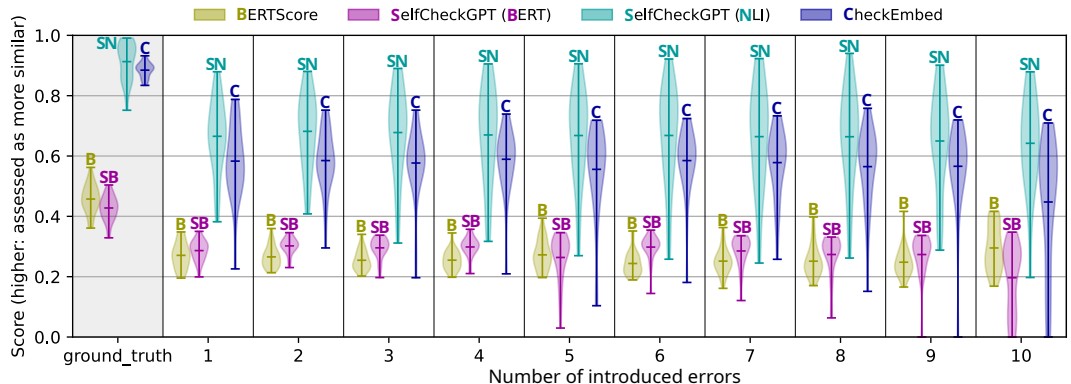

Figure 5: **Analysis of fine-grained hallucination verification of LLM answers (GPT-4o) when summarizing scientific documents**, details explained in Section 4.4.

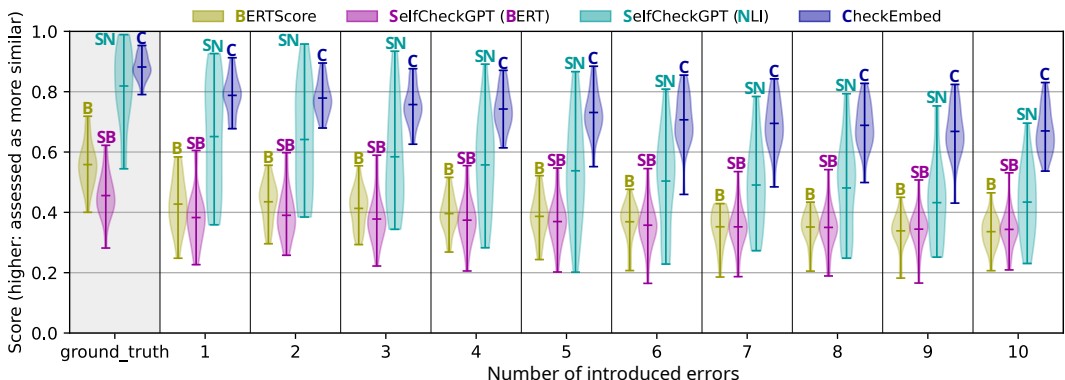

Figure 6: **Analysis of fine-grained hallucination verification of LLM answers (GPT-4o) when summarizing legal documents**, details explained in Section 4.4.

LLM to summarize these documents, while forcing deliberate small fact-level mistakes, from 1 to 10 mistakes per summary. CHECKEMBED is able to recognize when samples contains no errors, as illustrated by very large scores for GT. Moreover, interestingly, it can also recognize hallucinations after introducing a single error, as visible by no overlap between the GT and the consecutive data points. Finally, we can observe that the amount of low-confidence scores is somewhat increasing with the growing number of introduced errors. However, this increase only starts to be distinctive beyond 5 errors. The trends for BERTScore and SelfCheckGPT are similar, which illustrates that these baselines perform well for their intended use case.

## 4.5 ENSURING FAST PROCESSING & SCALABILITY

We also investigate the running times of all considered baselines. Example results are in Figure 7. The numbers for each datapoint correspond to the total runtime required to construct 20 embeddings and to compute similarity scores between all embedding pairs. We show runtimes for CHECKEM-BED with the Stella models as their smaller model sizes (435M, 1.5B) are comparable to the best available bidirectional embedding models that can be used with BERTScore and SelfCheckGPT (e.g., microsoft/deberta-xlarge-mnli has 750M parameters). CHECKEMBED , while using the Stella models, maintains a constant evaluation time regardless of the sample size or token number for the text chunks. All comparison baselines exhibit an inflation of their runtime, as we increase the number of samples or the token length of the inputs, making CHECKEMBED $30\times - 300\times$ faster. We present additional results for GPT and other embedding models in Appendix A.4.2. These results further showcases the high performance of CHECKEMBED, rooted in its simplicity: *all that is required to compute is a single embedding of a textual answer or its chunk.*

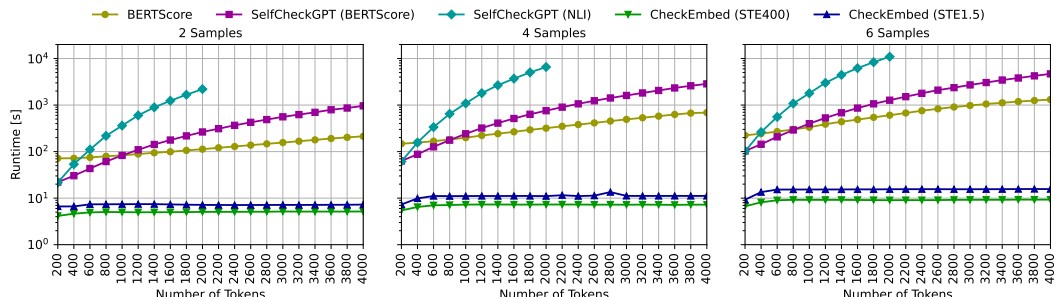

Figure 7: **Comparison of running times of CHECKEMBED and other baselines while varying the number of samples per datapoint.** We used an NVIDIA RTX3090 GPU for this experiment. Please note the logscale y axis.

## 4.6 ABLATION STUDY

Finally, we also look how the accuracy of CHECKEMBED is influenced by the sample size per datapoint. We conducted this evaluation on the WikiBio dataset and plot the Spearman's rank correlation coefficient while varying the number of samples in Figure 8. While all embedding models show an accuracy increase with more samples, the accuracy starts to stabilize with 8 samples (6 samples for SFR and E5), at which point the gain from using additional samples might be offset by the additional cost.

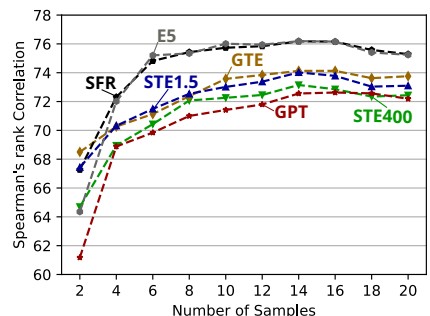

Figure 8: **Comparison of the accuracy of CHECKEMBED with different embedding models while varying the number of samples per datapoint.**

## 5 RELATED WORK

Trustworthy AI is a broad research area focusing on the transparency, fairness, and reliability of AI systems. Efforts in this field aim to develop frameworks and guidelines that ensure AI systems are trustworthy and align with human values (Huang et al., 2024; Liu et al., 2024). Initiatives like differential privacy (Behnia et al., 2022), fairness constraints in machine learning models (Jui & Rivas, 2024), and transparent reporting of AI capabilities and limitations (Liao & Vaughan, 2023) are prominent in this context. These approaches strive to build AI systems that are not only effective, but also ethically sound and socially acceptable.

Explainable AI (XAI) (Longo et al., 2024) is another critical area of research with the goal of making AI systems more transparent and interpretable to users. Several works have developed methods to enhance explainability in AI systems (Zhao et al., 2024a; Luo & Specia, 2024). For instance, self-explaining models that generate explanations alongside predictions have been explored to improve user trust and understanding (Huang et al., 2023b; Madsen et al., 2024). Other approaches include post-hoc explanation methods, which provide insights into model decisions after predictions are made, thus facilitating better human-AI interaction (Vale et al., 2022; Kroeger et al., 2024). These advancements are crucial for deploying AI in sensitive areas where understanding the rationale behind decisions is imperative.

The rise of AI has also prompted methodological and epistemological inquiries. Researchers are examining the foundational questions regarding how AI systems generate knowledge and the implications of these processes (Fleisher, 2022). Discussions in this domain focus on the nature of machine learning (Shanahan, 2023), the validity of AI-generated knowledge (Mahowald et al., 2024), and the ethical considerations surrounding AI deployment (Li, 2023; Radanliev & Santos, 2023). These inquiries are essential for framing the theoretical underpinnings of AI and addressing concerns related to bias, fairness, and accountability in AI systems.

The problem of hallucinations in LLMs has gathered significant attention (Rawte et al., 2023; Zhang et al., 2023b; Huang et al., 2023a; Ji et al., 2023; Bai et al., 2024). Chrysostomou et al. (2024) find that hallucinations are less prevalent in pruned LLM for summarization tasks, which they attribute

to an increased dependence on the original source. Various methods on detecting hallucation have been proposed, including SelfCheckGPT (Manakul et al., 2023), fact checking (Zhang et al., 2024a; Chern et al., 2023) and others (Su et al., 2024; Zhang et al., 2023a; Shi et al., 2023). Another focus is the reduction of hallucinations. Ever (Kang et al., 2024) dynamically verifies generated content against evidence during the generation process. Zhang et al. (2024b) propose the use of the human user and knowledge bases to align their knowledge to let the LLM answer truthfully. One of the goals of Retrieval Augemented Generation (RAG) (Zhu et al., 2024a) has been hallucination reduction by fetching relevant information for the LLM context. Benchmark efforts have also been proposed (Li et al., 2023a; Zhu et al., 2024b; Sun et al., 2024). We do not compare CHECKEMBED to schemes like MIND (Su et al., 2024), BARTScore (Yuan et al., 2021), UniEval (Zhong et al., 2022), or G-Eval (Liu et al., 2023) because their focuses differ from hallucination detection. MIND analyzes internal LLM states, which are often unavailable (we focus on simplicity); BARTScore evaluates text generation on multiple factors, with only one being loosely related to hallucinations; UniEval and G-Eval, while focused on text generation quality, do not center on detecting hallucinations as their primary goal.

LLM-based agents represent a burgeoning area (Xi et al., 2023), where LLMs are utilized as autonomous agents to perform complex tasks. These agents leverage the generative capabilities of LLMs to interact with users, perform tasks, and make decisions, often resorting to different prompt engineering techniques (Wei et al., 2023; Long, 2023; Yao et al., 2023; Besta et al., 2024a; Wang et al., 2023; Qiao et al., 2023; Besta et al., 2024b). Recent studies focus on enhancing the autonomy and effectiveness of these agents by improving their ability to understand and respond to nuanced user inputs (Barua, 2024). Techniques such as fine-tuning on specific tasks (Chen et al., 2024) and incorporating external knowledge sources (Guan et al., 2024; Liu et al., 2022) are employed to enhance the performance of LLM-based agents in real-world applications.

Finally, evaluating LLMs is an ongoing challenge given their complexity and the diverse range of tasks they can perform (Zhao et al., 2024b; Minaee et al., 2024). Traditional evaluation metrics often fall short in capturing the full spectrum of LLM capabilities. Hence, researchers are developing new benchmarks and evaluation frameworks that better reflect real-world use cases (Chang et al., 2024). These include task-specific evaluations, user-centric assessments (Wang et al., 2024a), and adversarial testing (Radharapu et al., 2023; Xu et al., 2024) to ensure that LLMs perform reliably across different scenarios and are resilient to manipulation.

## 6 CONCLUSION

Large Language Models (LLMs) are revolutionizing various domains, yet effective verification for open-ended tasks remains a significant challenge. Established methods, which focus on token- and sentence-level analysis, fall short in scalability and effectiveness. Addressing this gap is crucial as applications of LLMs expand, necessitating robust mechanisms to ensure the accuracy and reliability of their outputs.

To this end, we introduce CHECKEMBED, a scalable approach to LLM verification. CHECKEMBED leverages the effectiveness of answer-level embeddings to compare LLM answers with one another and the potential ground-truth. By transforming complex textual answers into individual embeddings using modern decoder-only based models like GPT Text Embedding Large, CHECKEMBED makes the verification process simple, accurate, and scalable. This straightforward methodology integrates seamlessly with modern data analytics infrastructure, highlighting its practical applicability and ease of deployment.

CHECKEMBED comes with a comprehensive verification pipeline that includes metrics and tools for assessing the veracity of LLM answers, such as heatmaps of similarites between embeddings of answers, the ground-truth, and statistical summaries. These tools provide detailed insights into the quality of LLM outputs and facilitate practical decision-making in real-world deployments. The simplicity of our approach allows for the extension of these metrics to various other applications, further enhancing its utility and flexibility.

Our pipeline has been tested on document analysis tasks, including term extraction. The results demonstrated significant improvements in accuracy and runtime performance compared to existing methods such as BERTScore (Zhang et al., 2020) and SelfCheckGPT (Manakul et al., 2023). These findings underscore the potential of CHECKEMBED to transform LLM verification in industrial settings, ensuring that LLM outputs are both reliable and scalable.

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

# A  APPENDIX / SUPPLEMENTAL MATERIAL

## A.1  PROMPTS

Table 2: Prompt template used for the query generation of the "similar description" use case. A list of "Generic" and "Precise" topics is used to replace ### HERE ### with an actual topic. The aim is to generate two passages of text that look different, but are the same content-wise.

---

### INSTRUCTION ###

Hello. Please generate two passages of text. They should both describe the same thing (### HERE ###). However, these two passages should differ VASTLY in their length, style.
I want you to give an answer using the following format:
<formatting>
### DESCRIPTION 1 ###
the actual description here...
### DESCRIPTION 2 ###
the actual description here...
</formatting>

### ANSWER ###

---

Table 3: Prompt template used for the query generation of the "different description" use case. A list of different topics is used to replace ### HERE 1 ### and ### HERE 2 ### with two actual topics. The aim is to generate two passages of text that seem alike, but are completely different content-wise.

---

### INSTRUCTION ###

Hello. Please generate two passages of text. They should describe two different things:
1. ### HERE 1 ###
2. ### HERE 2 ###

However, these two passages should have the same length and style.
I want you to give an answer using the following format:
<formatting>
### DESCRIPTION 1 ###
the actual description here...
### DESCRIPTION 2 ###
the actual description here...
</formatting>

### ANSWER ###

---

Table 4: Prompt template used to "extract respective terms and their definitions" from chunks of legal documentation. Given the complexity of the task, we provide the concrete format as well as an in-context example. [### REPLACE WITH CONTEXT ###] gets replaced by a text chunk from the legal definitions dataset.

---

### INSTRUCTION ###

You are a lawyer.

### QUESTION ###

Based on the provided context extract all the legal definitions. Answer using the following formatting.
<formatting>
Term.Definition
Term.Definition
...
</formatting>
<example>
[...]
### CONTEXT ###

Preliminary Note
The Stock Purchase Agreement sets forth the basic terms of the purchase and sale of the preferred stock to the investors (such as the purchase price, closing date, conditions to closing) and identifies the other financing documents. Generally this agreement does not set forth either (1) the characteristics of the stock being sold (which are defined in the Certificate of Incorporation) or (2) the relationship among the parties after the closing, such as registration rights, rights of first refusal and co-sale and voting arrangements (these matters often implicate persons other than just the Company and the investors in this round of financing and are usually embodied in separate agreements to which those others persons are parties, or in some cases in the Certificate of Incorporation). The main items of negotiation in the Stock Purchase Agreement are therefore the price and number of shares being sold, the representations and warranties that the Company must make to the investors and the closing conditions for the transaction.
SERIES A PREFERRED STOCK PURCHASE AGREEMENT
THIS SERIES A PREFERRED STOCK PURCHASE AGREEMENT (this "Agreement"), is made as of [], 20[], by and among [______], a Delaware corporation (the "Company"), and the investors listed on Exhibit A attached to this Agreement (each a "Purchaser" and together the "Purchasers").
The parties hereby agree as follows:

### ANSWER ###

Agreement. THIS SERIES A PREFERRED STOCK PURCHASE AGREEMENT
Company. Delaware corporation
Purchaser. Company or the investors listed on Exhibit A
Purchasers. Company and the investors listed on Exhibit A together
</example>

### CONTEXT ###

[###REPLACE WITH CONTEXT###]

### ANSWER ###

---

Table 5: Prompt template used for the ground-truth generation query of the "hallucination test" use case. A list of mostly scientific topic is used to replace ### TOPIC ###.

---

### INSTRUCTION ###

Hello. Please generate a passage of text that talks about (### TOPIC ###).

Please, use the following format for answering:
<formatting>
### PASSAGE ###
The passage here....
</formatting>

---

Table 6: Prompt template used for the hallucination generation query of the "hallucination test " use case. A list of mostly scientific topic is used to replace ### TOPIC ###. ### NUMBER ### is replaced according to an user-specified range of numbers. ### ERRORS ### is used during the hallucination generation process, but is removed from the sample output before the embeddings are created.

---

### INSTRUCTION ###

Hello. Please generate ### NUMBER ### completely false information (fact hallucinations) on (### TOPIC ###).
Then insert the errors inside a passage of text that talks about (### TOPIC ###).
You should convince a reader that the false information are actually correct ones.

Please, use the following format for answering:

<formatting>
### ERRORS ###
List of fact hallucinations to be later included in the passage...
### PASSAGE ###
The passage here....
</formatting>

---

## A.2 EMBEDDING LENGTH AND PARAMETER SIZE

Table 7: Embedding length and number of parameters for each model used during the evaluation.

| Model Name | Length | #Parameters |
|---|---|---|
| GPT Text Embedding Large | 3072 | not public |
| Salesforce/SFR-Embedding-Mistral | 4096 | 7.11B |
| intfloat/e5-mistral-7b-instruct | 4096 | 7.11B |
| Alibaba-NLP/gte-Qwen1.5-7B-instruct | 4096 | 7.72B |
| dunzhang/stella_en_1.5B_v5 | 4096 | 1.54B |
| dunzhang/stella_en_400M_v5 | 4096 | 435M |
| microsoft/deberta-xlarge-mnli | 1024 | 750M |
| roberta-large | 1024 | 355M |

## A.3 COMPUTE RESOURCES

Running the pipeline for the dataset of legal definitions for three LLMs (GPT-3.5, GPT-4 and GPT-4o as well as the baselines SelfCheckGPT and BERTScore) on a single NVIDIA Tesla V100-PCIE-32GB GPU took roughly 90 minutes. That dataset was used to create the heatmap figures 4 and 9. The pipeline for the datasets with similar and different descriptions, used for the violin plots, was executed on the same hardware in around 80 minutes. The experiments for the runtime comparison took 43 hours respectively for each GPU (NVIDIA A100 and NVIDIA RTX3090).

1026
1027

## A.4 ADDITIONAL RESULTS

### A.4.1 HEATMAPS

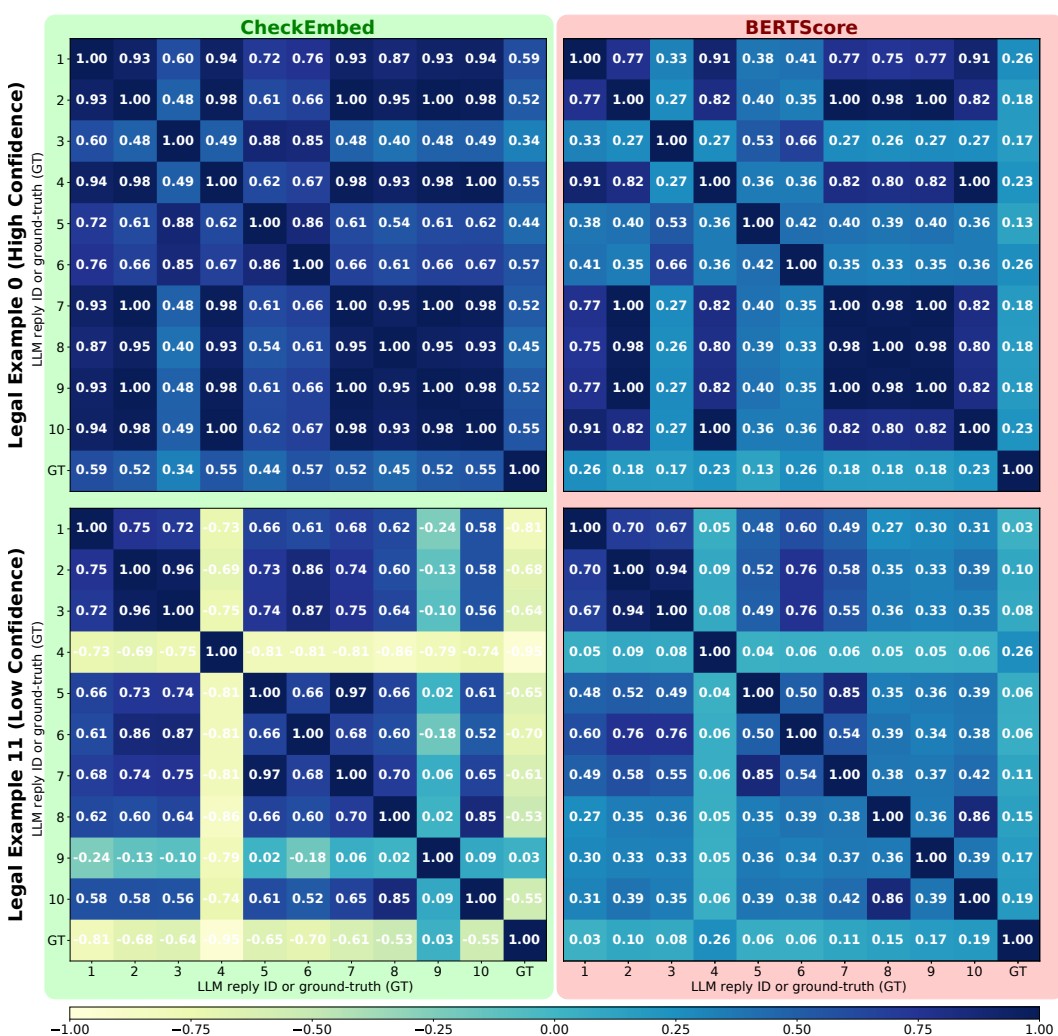

Figure 9: **Analysis of the verification of LLM answers (GPT-3.5)**, details explained in Section 4.2. We compare to BERTScore; Self-CheckGPT comes with significantly higher runtimes (detailed in Section 4.5) and less competitive scores as it does not focus on open-ended answer-level analysis. The results form a heatmap of the CHECKEMBED's, or BERTScore's, cosine similarity between all LLM replies, and between each reply and the human expert prepared ground-truth (GT). Rows correspond to two representative legal documents, that come with – respectively – high and low LLM confidence in its replies. Embedding model used in both rows: GPT Text Embedding Large.

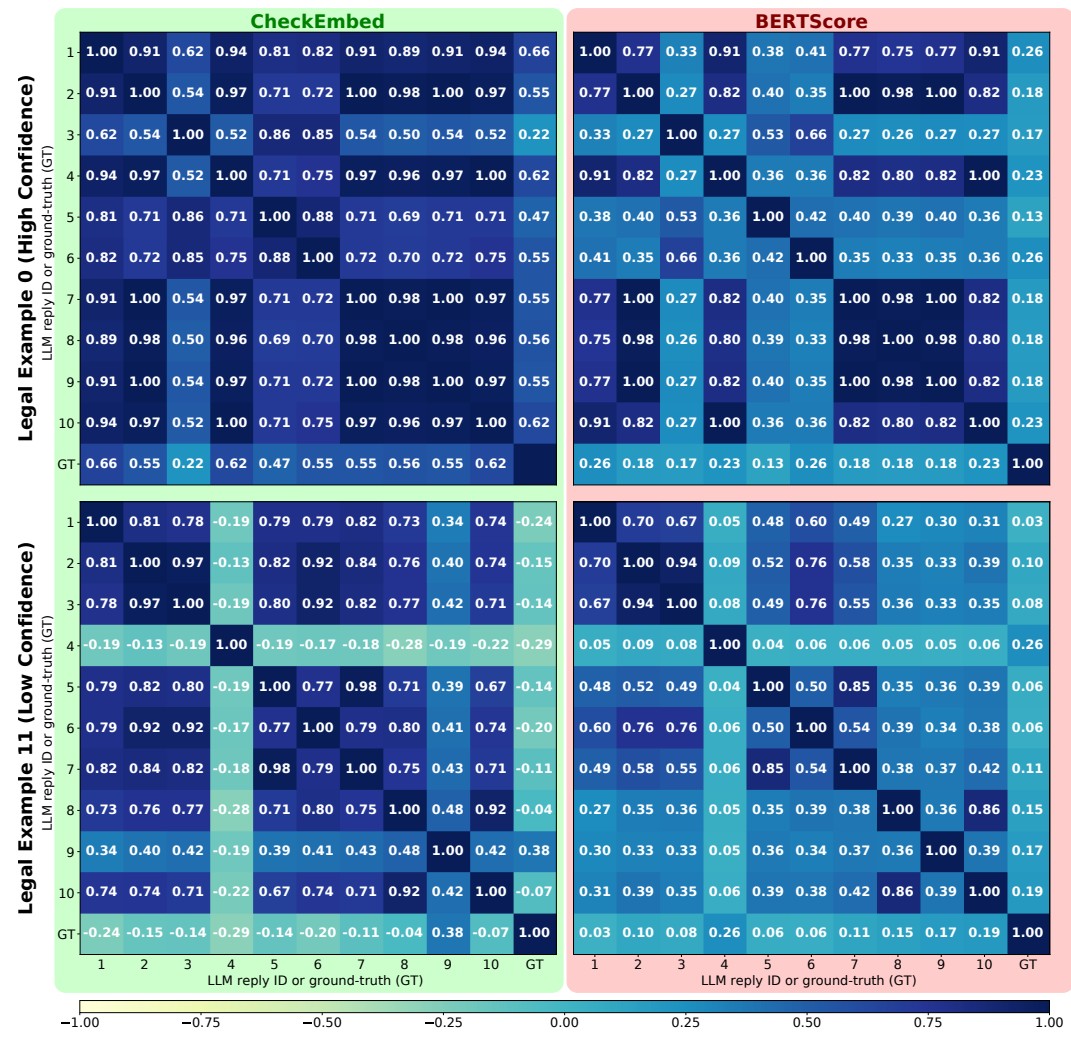

Figure 10: **Analysis of the verification of LLM answers (GPT-3.5)**, details explained in Section 4.2. We compare to BERTScore; Self-CheckGPT comes with significantly higher runtimes (detailed in Section 4.5) and less competitive scores as it does not focus on open-ended answer-level analysis. The results form a heatmap of the CHECKEMBED's, or BERTScore's, cosine similarity between all LLM replies, and between each reply and the human expert prepared ground-truth (GT). Rows correspond to two representative legal documents, that come with – respectively – high and low LLM confidence in its replies. Embedding model used in both rows: Stella 1.5B.

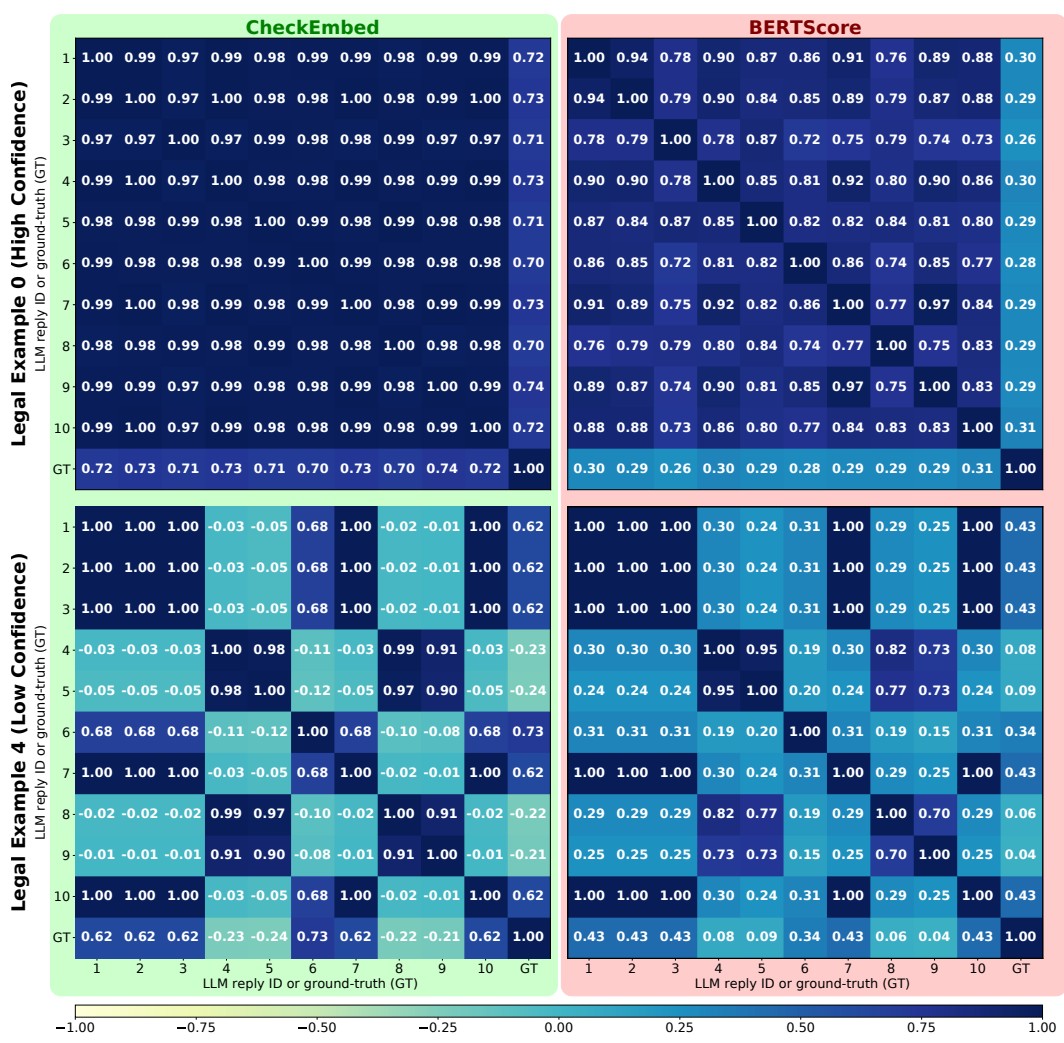

Figure 11: **Analysis of the verification of LLM answers (GPT-4)**, details explained in Section 4.2. We compare to BERTScore; Self-CheckGPT comes with significantly higher runtimes (detailed in Section 4.5) and less competitive scores as it does not focus on open-ended answer-level analysis. The results form a heatmap of the CHECKEMBED's, or BERTScore's, cosine similarity between all LLM replies, and between each reply and the human expert prepared ground-truth (GT). Rows correspond to two representative legal documents, that come with – respectively – high and low LLM confidence in its replies. Embedding model used in both rows: Stella 1.5B.

## A.4.2 RUNTIMES

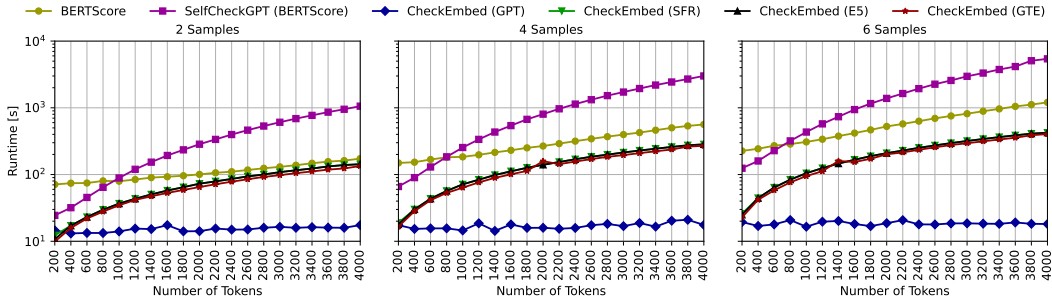

Figure 12: **Comparison of running times of CHECKEMBED and other baselines while varying the number of samples (2, 4 and 6) per datapoint.** We used an NVIDIA A100 GPU for this experiment. Please note the logscale y axis.

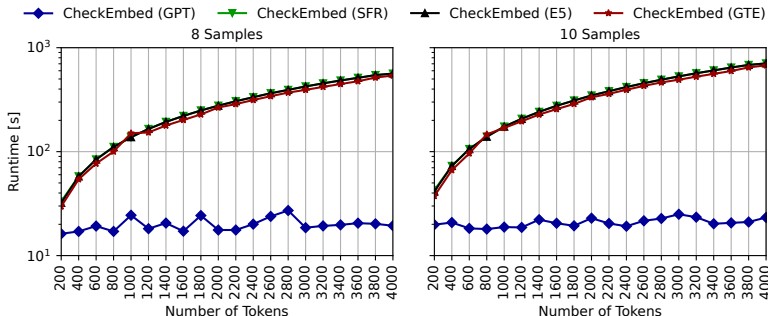

Figure 13: **Comparison of running times of CHECKEMBED and other baselines while varying the number of samples (8 and 10) per datapoint.** We used an NVIDIA A100 GPU for this experiment. Results for BERTScore and SelfCheckGPT (BERTScore) are missing, since their execution with larger sample sizes would have taken a long time. Please note the logscale y axis.

### A.4.3 FULL WIKIBIO RESULTS

Table 8: **CHECKEMBED results for the WikiBio benchmark.** PE stands for Pearson correlation coefficient and SP for Spearman's rank correlation coefficient.

| #Samples | SFR | | STE400 | | STE1.5 | | GPT | | E5 | | GTE | |
|---|---|---|---|---|---|---|---|---|---|---|---|---|
| | **PE** | **SP** | **PE** | **SP** | **PE** | **SP** | **PE** | **SP** | **PE** | **SP** | **PE** | **SP** |
| 2 | 61.9 | 67.3 | 59.7 | 64.7 | 62.2 | 67.4 | 52.3 | 61.2 | 59.9 | 64.4 | 63.8 | 68.5 |
| 4 | 67.9 | 72.3 | 64.4 | 68.9 | 66.3 | 70.3 | 63.1 | 68.9 | 68.8 | 72.0 | 67.8 | 70.3 |
| 6 | 70.6 | 74.8 | 66.5 | 70.4 | 68.4 | 71.5 | 64.6 | 69.8 | 71.9 | 75.2 | 69.3 | 71.1 |
| 8 | 71.0 | 75.4 | 67.4 | 72.1 | 68.9 | 72.5 | 65.0 | 71.0 | 72.4 | 75.3 | 70.0 | 72.4 |
| 10 | 71.6 | 75.7 | 68.2 | 72.3 | 69.5 | 73.0 | 65.6 | 71.4 | 73.3 | 76.0 | 71.0 | 73.6 |
| 12 | 71.2 | 75.8 | 67.7 | 72.5 | 69.2 | 73.4 | 66.0 | 71.8 | 72.9 | 75.9 | 71.2 | 73.8 |
| 14 | 71.7 | 76.2 | 68.0 | **73.1** | 69.5 | **74.0** | 66.5 | 72.6 | 73.2 | 76.2 | 71.4 | 74.1 |
| 16 | **72.2** | **76.2** | **68.5** | 72.9 | **69.9** | 73.8 | **66.8** | 72.6 | 73.6 | 76.2 | 71.6 | **74.1** |
| 18 | 71.4 | 75.6 | 67.7 | 72.3 | 69.2 | 73.0 | 66.7 | 72.6 | 72.9 | 75.4 | 71.0 | 73.6 |
| 20 | 71.5 | 75.3 | 68.0 | 72.4 | 69.6 | 73.1 | 66.7 | 72.2 | 72.9 | 75.2 | 71.3 | 73.8 |

