# OpenReview forum: "CheckEmbed: Effective Verification of LLM Solutions to Open-Ended Tasks"
_ICLR.cc/2025/Conference — Submitted to ICLR 2025_

### Official Review · Reviewer_2mmx · 2024-11-01

**Soundness:** 3
**Presentation:** 2
**Contribution:** 3
**Rating:** 3
**Confidence:** 3

**Summary:**

Accurately evaluating the performance of LLMs in open-ended question answering remains an unresolved challenge. This paper proposes a novel method, CheckEmbed, which leverages pre-trained embedding models to encode both sampled LLM responses and ground truths. The method calculates pairwise scores between these embedding vectors (e.g., cosine similarity). The resulting score matrix can be further processed to compute the Frobenius norm, representing the truthfulness and quality of the LLM responses. Experimental results demonstrate that CheckEmbed outperforms existing methods (such as BertScore and SelfCheckGPT) in both accuracy and efficiency.

**Strengths:**

- CheckEmbed is a novel method, which can more accurately identify and distinguish texts with similar or different semantics.
- CheckEmbed allows for cost-effective evaluation of LLMs on open-ended question answering, with the potential to become a widely used evaluation method.
- The research has a clear focus, the topic is of significant importance, and the work is well-completed.

**Weaknesses:**

- Some issues with the finer details of the paper. For instance, the two heatmaps in the main figure (Figure 2) have incorrect y-axis (or x-axis) labels.
- It seems that CheckEmbed does not outperform baseline methods in certain experiments, such as those in Sections 4.2 and 4.3.
- The paper's structure is somewhat disorganized. The experimental settings are only described textually, and the addition of formulas and examples would improve clarity. Specifically, the examples of similar/different replies and ground truth, the legal documents data point, and the detailed calculation process of Pearson correlation in Section 4.3 are needed.
- Some experimental analysis is lacking. For instance, the compared baselines are not LLM-based methods, and since embedding models are the most critical component of CheckEmbed, should there be an analysis of how different embedding models affect the results? Additionally, it would be useful to compare CheckEmbed against some existing LLM-based methods for evaluating open-ended QA, e.g., llm-as-a-judge.
- Intuitively, BertScore and SelfCheckGPT compare texts at a finer granularity (i.e., token- and sentence-level), while CheckEmbed operates at a passage-level, which is coarser. Finer-grained comparisons usually yield better results, yet the experimental results show otherwise. How can this counterintuitive outcome be explained? (e.g., is it due to differences in embedding models, etc.?)

**Questions:**

See "Weaknesses" section.

---

### Official Review · Reviewer_dz6R · 2024-11-03

**Soundness:** 2
**Presentation:** 3
**Contribution:** 2
**Rating:** 3
**Confidence:** 4

**Summary:**

This paper proposes a metric termed CheckEmbed which compares two replies based on their reply-level embeddings. The authors claim that their main contributions are the metric's effectiveness and strong performance on both constructed tasks and an existing benchmark, WikiBio. They argue that the effectiveness stems from its $O(1)$ computational complexity, compared to $O(n^2)$ in previous approaches, as NumPy supports efficient algorithms for computing cosine similarity or Pearson correlation. The strong performance is attributed to their use of paragraph-level comparison, in contrast to previous token-level comparison approaches.

**Strengths:**

I think comparing embeddings of replies (reply -> embed(reply)) should be much more effective than using GPT to compare two replies directly. If we can maintain performance while increasing preprocessing time, that would be ideal.

**Weaknesses:**

- The paper's main claim about effectiveness is questionable. If the performance improvements primarily stem from NumPy's implementation of faster algorithms for computing cosine or correlation scores, then the scientific contribution is overclaimed.
- The claim about paragraph-level comparison versus token-level comparison improving performance is suspicious. Lines 149-150 indicate that the main methods are cosine similarity and Pearson correlation between paragraphs. However, computing cosine similarity between two paragraph embeddings still essentially involves token-level comparison.
- The evaluation appears to be designed specifically to highlight flaws in BERTScore. The test setting using similar-but-different text is not sufficiently rigorous. While Figure 1 shows data where most tokens and patterns are similar with only a few different tokens, BERTScore would calculate many redundant token pairs, potentially overlooking key differences. However, realistic scenarios involve much more complex data patterns than these contrived examples. A more robust evaluation using data with sufficient pattern complexity should be added to prove the method's validity.
- The literature review is incomplete, leading to missing comparisons. In the related work section, 4 out of 5 paragraphs discuss distant areas or general domains while lacking scientific literature on evaluation metrics. The authors justify not comparing MIND, BARTScore, UniEval, or G-Eval by stating these metrics aren't designed for detecting hallucinations. However, this raises the question: was BERTScore specifically designed for this purpose? The boundaries between these metrics' applications are unclear. More discussion about the similarities and differences regarding their targets and detailed methods would be valuable.

**Questions:**

- Regarding Figure 2, why are 1/2 steps included in the Checkembed pipeline? Since Checkembed appears to be an evaluation module, batch inference features should be attributed to inference infrastructures like vLLM or SGLang.
- For lines 157-159, the claim of O(1) complexity requires stronger justification, especially since it seems independent of the embedding matrix size. This appears counterintuitive - could you elaborate on how this is achieved?
- "When calculating cosine similarity and Pearson correlation between k replies of varying lengths:
How are the different lengths handled? For Pearson correlation specifically, is comparing tokens in their natural order meaningful when the data shows similar patterns with only a few token substitutions?"

---

### Official Review · Reviewer_YESP · 2024-11-06

**Soundness:** 3
**Presentation:** 3
**Contribution:** 2
**Rating:** 3
**Confidence:** 4

**Summary:**

The paper introduces CheckEmbed, a framework for verifying LLM responses on open-ended generation tasks. At its core, CheckEmbed uses answer-level embeddings to calculate a single representation of an answer, comparing generated answers with ground-truth via cosine similarity. This core technique is integrated into a pipeline which helps to measure + explain the stability of LLM outputs, and use this to draw takeaways wrt to the correcteness of model outputs. The authors find CheckEmbed outperforms conventional similarity metrics such as BERTScore and SelfCheckGPT on tasks formulated to test whether the metrics can distinguish between 'similar' and 'different' replies (tested over different domains and granularities). They also observe this method (due to its simple answer-level embedding) scales much more efficiently than the two baseline methods.

**Strengths:**

The paper was generally well-written and easy to follow. The figures are generally very clear (in particular, Figure 2 was very helpful to quickly visualize the differences between approaches). The motivation seemed fairly clear -- namely, targeting a more efficient and accurate method for comparing generated texts. The technique largely leverages a simple embedding method, however, they explore methods for aggregating this technique through repeated LLM sampling.

**Weaknesses:**

- Narrow evaluation scope: Much of the evaluation is conducted on synthetic or in-house data, raising questions about the generalizability of the results. The strongest improvements are seen on these datasets, while for the WikiBio task, other metrics—such as the NLI variant of SelfCheckGPT—demonstrate comparable performance. A more extensive evaluation on public datasets would provide stronger evidence for the method’s general applicability.

- Lack of novelty: Generally, this seems like a straightforward application of holistic 'answer-level' embeddings. While the introduced pipeline affords exploration of practical use cases, more work is needed to conclusively evaluate the benefits of this on real-world tasks.

- Reliance on synthetic data: Section 4.1 uses synthetically generated data for the evaluation, which may limit the validity of broad claims drawn from this analysis. For instance, the specific generation prompts used could inherently favor the answer-level embedding technique. Given that this subtask is central to the CheckEmbed approach, additional real-world evaluation would be valuable for supporting the conclusions.

- Lack of sampling details: The paper lacks detail on specific LLM sampling parameters (e.g., temperature, top_k values), which could significantly impact the histogram-based approaches in CheckEmbed. This information seems pretty important, as sampling + aggregation is one of the key contributions, and the variability of LLM responses (due to sampling parameters) could significantly affect the resultant stability/confidence results.

- Insufficient evidence for specific claims: Some of the interesting observations, such as in Section 4.2 (e.g., "Figure 4 shows that whenever CheckEmbed has high confidence in the LLM replies, there is a high likelihood that these replies are close to the corresponding GT"), appear more like case studies than generalized findings. While it would be interesting if CheckEmbed’s inter-sample similarity correlates well with correctness, the two legal examples provided aren't convincing. Additional evidence across the entire legal dataset, for instance, would strengthen this claim.

**Questions:**

- Using histogram to do thresholding: Have you explored any methods for automatically deriving 'useful' confidence thresholds?

- Sampling methods: What sampling methods did you use/explore when generating LLM responses, and how might these impact your results? This is a pretty crucial detail which could effect your pipeline performance.

- Proprietary Datasets: Can you provide additional details on the proprietary and synthetic datasets used in evaluation? E.g. how many examples were these results computed over? Can you release the synthetic datasets produced?

- LLM-as-Judge Comparison: How does CheckEmbed’s embedding-based verification approach compare to LLM-as-judge methods for answer verification? Do you have comparisons with any existing LLM baselines?

Typos / Presentation Suggestions

- l248: noticely
- ll356 – Consistently
- Figure 5 and Figure 6 – can you add trendlines to these?
- Figure 9/10/11 – can you clearly outline the different parameters at the beginning for better readability? (e.g. [GPT-4 sampling, Stella 1.5B embeddings])

---

### Official Review · Reviewer_6bnj · 2024-11-07

**Soundness:** 1
**Presentation:** 2
**Contribution:** 1
**Rating:** 3
**Confidence:** 4

**Summary:**

This paper addresses LLM hallucination detection and answer verification. Following the assumption of SelfCheckGPT, the authors propose sampling multiple responses from an LLM using the same query, where semantic dissimilarity among responses indicates potential hallucination.
The paper introduces CHECKEMBED, a method for verifying LLM outputs in open-ended tasks. The authors claim three main contributions: 1) utilizing response-level embeddings rather than token or sentence-level comparisons, achieving faster processing than BERTScore and SelfCheckGPT; 2) developing a comprehensive verification pipeline with embedding similarity heatmaps and statistical summaries; 3) validating the approach on document analysis and WikiBio generation tasks. Experimental results suggest improvements in both accuracy and runtime compared to existing methods.

**Strengths:**

The paper addresses a crucial challenge in verification of open-ended LLM tasks

The proposed method is straightforward and practical to implement

**Weaknesses:**

1. Despite claiming to verify open-ended tasks, the method effectively only works for hallucination detection tasks with definitive answers. Multiple divergent responses could all be valid for truly open-ended queries (e.g., "Tell me a joke"), which the method fails to accommodate.

2. Regarding Contributions 1 and 3, computing cosine similarity between text embeddings is a well-established approach. BERTScore and SelfCheckGPT intentionally utilize token-level and sentence-level information to overcome the limitations of direct embedding similarity. CHECKEMBED's regression to embedding similarity, while claiming superiority over more granular methods, requires validation on public, representative text similarity or QA-Eval tasks. The simple embedding-based approach may overlook complex logical relationships and reasoning errors, particularly in specialized domains (legal, medical), and fails to consider contextual coherence or detect plausible but contradictory content.

3. The legal document analysis experiments are limited (only two cases shown), lacking statistical significance tests to support performance claims. While the WikiBio experiments provide some representativeness, deeper analysis is needed to determine whether performance gains stem from the CHECKEMBED framework or the GTE model, as BERTScore and SelfCheckGPT could potentially be enhanced with GTE.

4. Regarding Contribution 2, given that "stability" was introduced by SelfCheckGPT and merely applied here, the claimed "comprehensive verification pipeline" lacks innovative components.

5. Contribution 4's efficiency comparisons hold limited value without addressing the aforementioned concerns.

6. The paper lacks sufficient discussion distinguishing itself from other embedding-based approaches.

**Questions:**

See weaknesses

---

### Meta-Review · Area_Chair_PyLi · 2024-12-18

**Metareview:**

**Summary:**

This work highlights the limitations of SelfCheckGPT and BERTScore, which are models designed to evaluate the similarity between two passages. To address these shortcomings, it introduces CheckEmbed, a novel approach that evaluates the embeddings of entire LLM-generated responses or large chunks of text, rather than focusing solely on individual sentences, facts, or tokens. Experimental results demonstrate that CheckEmbed achieves significant improvements in accuracy, cost-effectiveness, and runtime performance compared to BERTScore and SelfCheckGPT.

**Strength:**
* The paper provides a clear motivation for targeting a more efficient and accurate method for comparing generated texts.
* CheckEmbed enables cost-effective evaluation of LLMs on open-ended question-answering tasks, with potential for broad adoption.
* The research addresses an important topic with a clear and focused approach, resulting in well-completed work.

**Weaknesses:**
*  Contributions 1 and 3 largely regress to well-known cosine similarity approaches, which are less innovative compared to token- or sentence-level methods like BERTScore or SelfCheckGPT.
* The approach requires validation on standard text similarity or QA evaluation tasks to substantiate claims of superiority over more granular methods.
* Heavy reliance on synthetic or in-house data limits generalizability. Public dataset evaluations are minimal, and comparable performance by other metrics (e.g., NLI variant of SelfCheckGPT) raises questions about claimed improvements.

**Additional Comments On Reviewer Discussion:**

The authors chose not to participate in the rebuttal process.

---

### Decision · Program_Chairs · 2025-01-22

Reject